# Effect of Pressure on the Superconducting Properties of Tl$_2$Ba$_2$Ca$_2$Cu$_3$O$_{9-\delta}$

**Anota O. Ijaduola [1],\*, Rai Shipra [2] and Athena S. Sefat [2]**

[1]    Department of Physics, University of North Georgia, Dahlonega, GA 30597, USA
[2]    Materials Science and Technology Division, Oak Ridge National Laboratory, Oak Ridge, TN 37831, USA;
       asake98@yahoo.com (R.S.); sefata@ornl.gov (A.S.S.)
\*    Correspondence: anota.ijaduola@ung.edu; Tel.: +01-706-867-2002

**Abstract:** This study investigated the application of pressure on the superconducting properties of a thallium-based cuprate, namely Tl$_2$Ba$_2$Ca$_2$Cu$_3$O$_{9-\delta}$ (Tl-2223). The superconducting transition temperature ($T_c$) and the critical current density ($J_c$) were studied by applying ~1 GPa of pressure. This hydrostatic pressure was applied in a piston-cylinder-cell (PCC), using Pb as a manometer and Daphne 7373 oil as the pressure transmitting medium. For estimating the $J_c$, we used Bean's critical state formula on the magnetic hysteresis curves at 10 K and 20 K. Both the $T_c$ and $J_c$ improved with pressure. The $J_c$ values increased at both temperatures and the $T_c$ value increased by 4 K with a pressure of 0.8 GPa. These results clearly indicate that pressure is another tool to control properties of quantum materials.

**Keywords:** cuprates; Tl-2223; pressure

## 1. Introduction

Despite the great progress made in the last years on expanding the fundamental and practical knowledge about superconductors, we are yet to fully understand the underlying microscopic mechanism responsible for high temperature superconductivity. In cuprates, the state of superconductivity emerges through the suppression of magnetism in the undoped materials, which are called "parents". The superconducting state in a parent is produced predominantly through chemical substitution, but may also be invoked by high pressure. The pressure alters the lattice constants through bond lengths and angles, which inevitably affect the electronic and magnetic correlations. However, the role of pressure may be complicated by the level of hydrostaticity, as uniaxial pressures on non-cubic systems may result in much different properties compared to isotropic pressures [1].

Among all Tl-based cuprate superconductors, Tl$_2$Ba$_2$Ca$_2$Cu$_3$O$_{10+\delta}$ (Tl-2223) has the highest transition temperature ($T_c$ ~ 128 K) and a better phase stability during synthesis at ambient pressure, for example compared to TlBa$_2$Ca$_2$Cu$_3$O$_{9-\delta}$ (Tl-1223) with a similarly high $T_c$ (~123 K) [2,3]; moreover, it has very efficient flux pinning properties. These features make Tl-2223 one of the preferable candidate superconductors for fundamental and applied research; hence, it is the focus of study here. We applied pressure on an already high-$T_c$ Tl-2223 in the hopes of achieving better superconducting properties. For these polycrystalline samples, we chose to apply an approximately uniform pressure using the piston-cylinder-cell (PCC) technique, while using Daphne 7373 as the pressure-transmitting medium. This PCC was designed solely for use with Quantum Design Magnetic Property Measurement System (MPMS), with the factory specification of a maximum reachable pressure of 1.3 GPa. The pressure applied to the sample was quasi hydrostatic and was applied from all directions, since the sample was fully immersed in the oil. Although the pressure dependence of $T_c$ in Tl-2223 has been investigated in previous works [4–7], the focus of this study was to add knowledge regarding the effect of pressure

on the inductively (magnetically) derived critical current density ($J_c$). Both works from D. T. Jover et al. [5,6] found an initial increase of 1.75K/GPa yielding a maximum $T_c$ of 133 K at a pressure of 4.2 GPa.

## 2. Materials and Methods

Crystalline samples were synthesized from a precursor having a composition equivalent to $Ba_2Ca_2Cu_3O_7$, and then, an appropriate molar amount of $Tl_2O_3$ was added to complete the formation of the desired compound. (See [8] for more details on the synthesis). Our sample measured 2.67 ± 0.05 mm by 1.28 ± 0.05 mm and was 1.04 ± 0.05 mm thick. The sample (with pure lead as the manometer) was loaded inside the PCC which was then mounted in the MPMS. The PCC was a non-magnetic beryllium copper cylindrical cell with up to a 2.6 mm diameter high pressure region. It used a Teflon sample tube and Teflon caps to form the high pressure seals. This insured that the sample moved relatively easily with little risk of damage to any of the pressure cell components. Pressure was generated in the Teflon tube filled with the Daphne 7373 oil. At low temperatures, the pressure was determined from the superconducting transition temperature of the pure lead. We then used the standard $dT_c/dP$ value for pure lead to calculate the corresponding change in sample pressure.

Our magnetometer had a 7 T superconducting magnet. In order to determine the superconducting transition temperature $T_c$, we collected zero-field-cooled (ZFC) data while increasing the temperature from 5 K to 150 K and the field-cooled-data (FC) while cooling from 150 K in an applied magnetic field of 20 Oe. The $J_c$ values were inductively (magnetically) determined by applying the modified critical state model [9,10] to the magnetic hysteresis produced by a rectangular sample with sides $b > a$ via the relation:

$$J_c = \frac{20\Delta M}{a\left(1 - \frac{a}{3b}\right)} \tag{1}$$

Here, $\Delta M = M^- - M^+$ is the magnetic hysteresis, where $M^-$ ($M^+$) is the magnetization at temperatures $T$ measured in decreasing (increasing) field $H$ history. Fields in the range 0–5.5 T were applied at two different fixed temperatures (10 K and 20 K) and the moment generated by the induced flowing current in the crystal was measured. Before beginning a measurement, the magnet was reset to eliminate any trapped flux and ensure that $H$ was truly zero.

## 3. Results and Discussions

Figure 1a,b are the temperature-dependent magnetic moment of the Tl-2223 sample at ambient pressure (open symbols) and an applied pressure of 0.8 ± 0.01 GPa (closed symbols) at ZFC and FC conditions, respectively. Both plots show an increase in the $T_c$ of the sample. In order to see this change in $T_c$ better, we show the zoomed-in view around the transition temperature of the ZFC curves in Figure 1c. The $T_c$ increased by 4 K (from 117 ± 0.04 K to 121 ± 0.04 K) with the applied pressure. This was in agreement with earlier works [4–6], where an increase of only 5 K at an applied pressure of 1.3 GPa was reported. This made us believe that this superconducting property will not have a drastic increase for this material even at high pressure.

Figure 2 portrays a typical magnetization hysteresis loop (MHL) of the Tl-2223 sample at $T = 10$ K and zero (ambient) applied pressure with applied magnetic fields up to 4.6 T. The shape of the MLH is highly symmetric as is typical of most cuprates. The magnetic fields are first applied in the increasing direction so as to reduce de-magnetization effects. For clarity purposes, we also show how the $\Delta M = M^- - M^+$ that is used in the Beans model equation is defined.

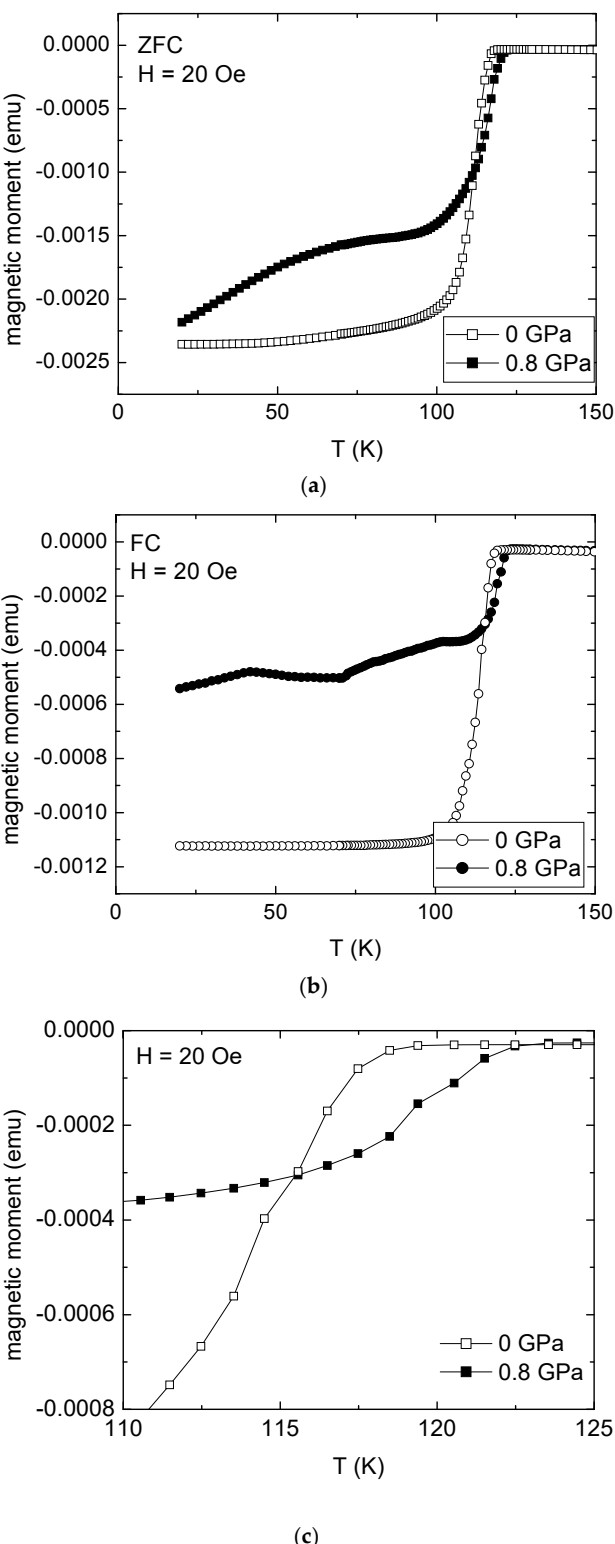

(**a**)

(**b**)

(**c**)

**Figure 1.** (**a**) Temperature-dependent magnetic moment results measured under zero-field-cooled (ZFC) condition and 20 Oe assuming perfect diamagnetism; (**b**) Temperature-dependent magnetic moment results measured under field-cooled-data (FC) condition and 20 Oe assuming perfect diamagnetism; (**c**) A zoomed-in view (around the transition temperature) of the ZFC curves of the temperature-dependent magnetic moment.

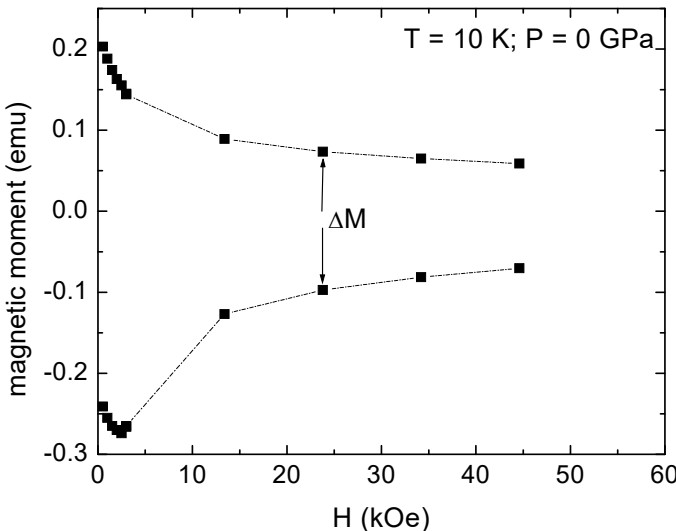

**Figure 2.** A typical magnetic hysteresis loop (MHL) of the Tl-2223 sample at *T* = 10 K and zero applied pressure.

We now consider the effect of pressure on the $J_c$ of the sample. To our knowledge, not much work has been done on the magnetically derived $J_c$ from Tl-2223. This could be attributed to the difficulties in obtaining good crystals and performing pressure measurements in general. We look at Figure 3a,b for quantitative details of the effect of pressure on the $J_c$ values at different applied magnetic fields. The calculated value of $J_c$ at *T* = 10 K and *H* = 1.5 kOe were 35,629 A/cm$^2$ and 37,448 A/cm$^2$ at zero and 0.8 GPa pressures, respectively. This signifies an increase of 5.1%. Similarly, the $J_c$ values at *H* = 3 kOe were 33,356 A/cm$^2$ and 346,689 A/cm$^2$ which is also about a 5% increase. This trend is similar to what we observed at the higher temperature of *T* = 20 K. Due to the low values of the $J_c$ in the high magnetic field range (from 13.4 kOe to 44.6 kOe), the percentage of increase were very high. For example, the $J_c$ at *H* = 13.4 kOe and *T* = 20 K are 4323 A/cm$^2$ and 4957 A/cm$^2$ at 0 and 0.8 GPa pressure, respectively. This is an increase of 14%. This number was even higher at *H* = 44.6 kOe. Thus, rather than computing the percentage increases, we concluded that there was an increase in the $J_c$ upon application of pressure. Lastly, from *H* = 50 kOe upwards, the $J_c$ was close to zero; this is the so-called "irreversibility field region" for superconductors, where the two MHLs (obtained in decreasing and increasing field *H* history respectively) collapse into a single one.

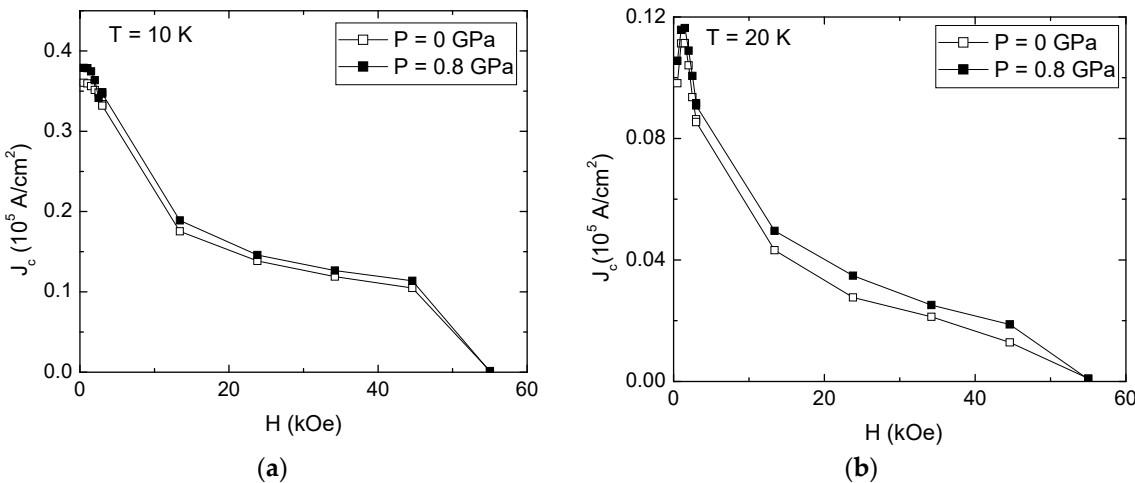

(a)　　　　　　　　　　　　　　　　　　　　　　　(b)

**Figure 3.** (**a**) Field dependence of the critical current density $J_c$ at *T* = 10 K for the Tl-2223 sample, calculated using the Bean's model; (**b**) Field dependence of the critical current density $J_c$ at *T* = 20 K for the Tl-2223 sample, calculated using the Bean's model.

## 4. Summary

We have studied the effect of pressure on the critical temperature $T_c$ and the critical current density $J_c$ of $Tl_2Ba_2Ca_2Cu_3O_{9-\delta}$ (Tl-2223) crystals. The pressure was applied in a piston-cylinder-cell using Pb as manometer and Daphne 7373 oil as the pressure transmitting medium. With the application of $0.8 \pm 0.01$ GPa pressure, the $T_c$ increased by 4 K. Additionally, we saw an increase in the $J_c$, which were measured via magnetic induction at temperatures of 10 K and 20 K. There was a 5% increase at low applied magnetic fields while the increase was about 14% at high applied magnetic fields. This percentage was higher due to the relatively low values of the $J_c$. These results show that pressure is another parameter that could be used to improve the superconducting properties.

**Author Contributions:** Conceptualization, A.S.S.; methodology, A.O.I. and A.S.S.; material preparation/resources, R.S.; formal analysis, A.O.I.; investigation, A.O.I. and A.S.S.; data curation, A.O.I.; writing—original draft preparation, A.O.I.; writing—review and editing, A.O.I. and A.S.S.; funding acquisition, A.S.S.

**Funding:** This research at Oak Ridge National Laboratory was funded by the U.S. Department of Energy, Basic Energy Sciences, Materials Sciences and Engineering Division.

**Acknowledgments:** This research was supported in part by an appointment to the Higher Education Research Experience for Faculty Program at Oak Ridge National Laboratory.

**Conflicts of Interest:** The authors declare no conflict of interest.

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
