# Peer review of "Effect of Pressure on the Superconducting Properties of Tl2Ba2Ca2Cu3O9-δ"

_crystals, doi:10.3390/cryst9010004_

Round 1

Reviewer 1 Report

Ijaduola et al. report the pressure effects (with applied pressure of 1 GPa) on superconducting transition temperature and critical current density in one of the Tl-based cuprates namely, Tl-2223. Although results are sound, before the publication several issues need to be addressed.

1) Authors use a piston type pressure cell and the pressure is checked at low temperatures using the Tc calibration of lead. On the other hand, transition temperature of the Tl-2223 is at considerably higher temperatures. How is the pressure gradient inside the cell with temperature?

2) A very sharp transition of the magnetization has been observed for the 0 GPa case (Fig 1). However, this transition is significantly broadened with a small amount of pressure. Can authors give an explanation?

3) What would be the error bars in critical current values?

Author Response

Dear Reviewer:

Thank you for taking the time to review our manuscript. Your suggestions for improvement are well taken, and have been incorporated in the revised edition. I have provided a detailed response to each of the issues that you raised, below.

Sincerely,

Anota Ijaduola

Reviewer's comments and responses:

1) Authors use a piston type pressure cell and the pressure is checked at low temperatures using the Tc calibration of lead. On the other hand, transition temperature of the Tl-2223 is at considerably higher temperatures. How is the pressure gradient inside the cell with temperature?

This is a valid concern. Use of lead (Pb) as reference in piston type pressure cell is very common, and a standard as noted in many publications, with a few cited in this work. The sample pressure is a function of temperature, but this dependence is negligible for temperature less than about 70 K. For higher temperature (close to or around room temperature), the manufacturers recommend using gadolinium which shows ferromagnetic ordering near 300 K. However, for this work, since the temperatures are not close to 300 K, the use of Pb is standard practice without jeopardizing the results/conclusions. The critical current density data were measured at 10 K and 20 K (well below 70 K), so the errors associated with the temperature dependence is even less of an issue there. The only concern is in the Tc measurement of the sample. A fair argument might be if we are sure that the pressure is up to the 0.8 GPa as stated, but the 4 K increase that we see in the Tc is in agreement with earlier works (please see Refs 4, 5, & 6). In summary, although the pressure will be different from the stated value due to the 120 K temperature (higher than the 70 K recommended by the manufacturers), the variation should not be significant enough to change the conclusion/result obtained in this work.

2) A very sharp transition of the magnetization has been observed for the 0 GPa case (Fig 1). However, this transition is significantly broadened with a small amount of pressure. Can authors give an explanation?

In most superconducting samples, due to their layered structures, the breadth of the transition has to do with the crystalline size distribution and the quality of the samples. It is not surprising that the transition is broadened with application of pressure because applying pressure affects the packing of crystallites within the polycrystalline piece (it is not as dense as a single crystal).  The very sharp transition that is observed in the 0 GPa case assures us that our sample is of a high quality. This is why pressure was applied to the same sample that was used in the 0 GPa case so we are sure that whatever difference(s) we see is due to the pressure effects onto a polycrystalline piece. In addition, the value of the Tc was read at the point of diamagnetic onset. This ensures that the length of the tail of the transition does not play a significant role.

3) What would be the error bars in critical current values?

      The critical currents were obtained from using the Bean’s model, (equation on line 70). The two variables in the equation are the magnetic moment and the dimensions of the samples. The SQUID magnetometer with its exceptional sensitivity and highly sophisticated software can resolve magnetic moment changes as small as 10-8 emu. Despite this, we still need a way to assess the quality of the data after it has been collected. The regression value provides this means. All of our data has a regression value that is close to 0.996 which implies that the raw data fits the expected dipole response. We used a vernier caliper with an error estimate of 0.05 mm to get the dimensions of the sample. This gives a calculated uncertainty of 0.0031 mm on the length dimension. This is very small compared to the range of values of the current density hence they are not shown on the graphs.

Reviewer 2 Report

This is a very well done work. The reported results are very interesting. The influence of pressure in the superconducting properties of a thallium-based cuprate has been accurately characterized. Experiments have been carried out using a piston-cylinder apparatus. They have been performed using state-of-the-art methods. The paper is well written and the conclusions are well supported. The authors do not leave room for critics. I only have a few minor recommendations.

Indicate the accuracy in determining pressure.

Explain how temperature was measured and give the estimated error.

Give a reference for the Beans model equation.

Rewrite abstract and summary highlighting your achievements.

Author Response

Dear Reviewer:

Thank you for taking the time to review our manuscript. Your suggestions for improvement are well taken and have been incorporated in the revised edition. I have provided a detailed response to each of the issues raised below.

Sincerely,

Anota Ijaduola

Reviewer's comments and responses:

1) Indicate the accuracy in determining pressure.

 The use of lead (Pb) as a reference in piston type pressure cell is very common, and a standard, as evidenced by the publications cited in this work. The sample pressure was inferred by noting the shift of superconducting transition temperature (Tc) of Pb with the cell compression. The main source of error will be in using the vernier caliper (with an error of 0.05 mm) to measure the cell compression. The calibration plot for our pressure cell shows a pressure of less than 0.01 GPa at a compression of 0.05 mm, so we estimate that our reported pressure should be accurate up to about 0.01 GPa. We have included this error estimate in all the pressure values in the manuscript.

2) Explain how temperature was measured and give the estimated error.

 The temperatures were measured by the SQUID magnetometer. They were displayed in the data file as a seven significant digit number which shows their high accuracy. We did not display all the significant digits in the graphs due to obvious reasons but from the type of results we present in this manuscript, we can give an estimated error of 0.04 K or lower. We have included this estimated error in the Tc values.

3) Give a reference for the Beans model equation.

 References 9 and 10 are for the Beans model equation. They are already included in the manuscript.

4) Rewrite abstract and summary highlighting your achievements.

Done. The abstract and the summary have been rewritten to better highlight the main results.